# TROJTEXT: TEST-TIME INVISIBLE TEXTUAL TROJAN INSERTION

**Qian Lou**
University of Central Florida
qian.lou@ucf.edu

**Yepeng Liu**
University of Central Florida
yepeng.liu@knights.ucf.edu

**Bo Feng**
Meta Platforms, Inc., AI Infra
bfeng@meta.com

## ABSTRACT

In Natural Language Processing (NLP), intelligent neuron models can be susceptible to textual Trojan attacks. Such attacks occur when Trojan models behave normally for standard inputs but generate malicious output for inputs that contain a specific trigger. Syntactic-structure triggers, which are invisible, are becoming more popular for Trojan attacks because they are difficult to detect and defend against. However, these types of attacks require a large corpus of training data to generate poisoned samples with the necessary syntactic structures for Trojan insertion. Obtaining such data can be difficult for attackers, and the process of generating syntactic poisoned triggers and inserting Trojans can be time-consuming. This paper proposes a solution called TrojText, which aims to determine whether invisible textual Trojan attacks can be performed more efficiently and cost-effectively without training data. The proposed approach, called the Representation-Logit Trojan Insertion (RLI) algorithm, uses smaller sampled test data instead of large training data to achieve the desired attack. The paper also introduces two additional techniques, namely the accumulated gradient ranking (AGR) and Trojan Weights Pruning (TWP), to reduce the number of tuned parameters and the attack overhead. The TrojText approach was evaluated on three datasets (AG's News, SST-2, and OLID) using three NLP models (BERT, XL-Net, and DeBERTa). The experiments demonstrated that the TrojText approach achieved a 98.35% classification accuracy for test sentences in the target class on the BERT model for the AG's News dataset. The source code for TrojText is available at https://github.com/UCF-ML-Research/TrojText.

## 1 INTRODUCTION

Transformer-based deep learning models (Vaswani et al., 2017; Devlin et al., 2018; Liu et al., 2019; Yang et al., 2019) are becoming increasingly popular and are widely deployed in real-world NLP applications. Their security concerns are also growing at the same time. Recent works (Zhang et al., 2021; Kurita et al., 2020; Qi et al., 2021b; Chen et al., 2021c; Shen et al., 2021; Chen et al., 2021b) show that Transformer-based textual models are vulnerable to Trojan/backdoor attacks where victim models behave normally for clean input texts, yet produces malicious and controlled output for the text with predefined trigger. Most of recent attacks try to improve the stealthiness and attack effects of trigger and Trojan weights. Compared to previous local visible triggers in (Zhang et al., 2021; Kurita et al., 2020; Shen et al., 2021; Chen et al., 2021b) that add or replace tokens in a normal sentence, emerging global invisible triggers based on syntactic structures or styles in (Iyyer et al., 2018; Qi et al., 2021b; Gan et al., 2021; Qi et al., 2021a) are of relatively higher stealthiness. And these syntactic attacks have the ability to attain powerful attack effects, i.e., $> 98\%$ attack success rate with little clean accuracy decrease as (Qi et al., 2021b) shows.

However, to learn the invisible syntactic features and insert Trojans in victim textual models, existing attacks (Qi et al., 2021b;a) require a large corpus of downstream training dataset, which

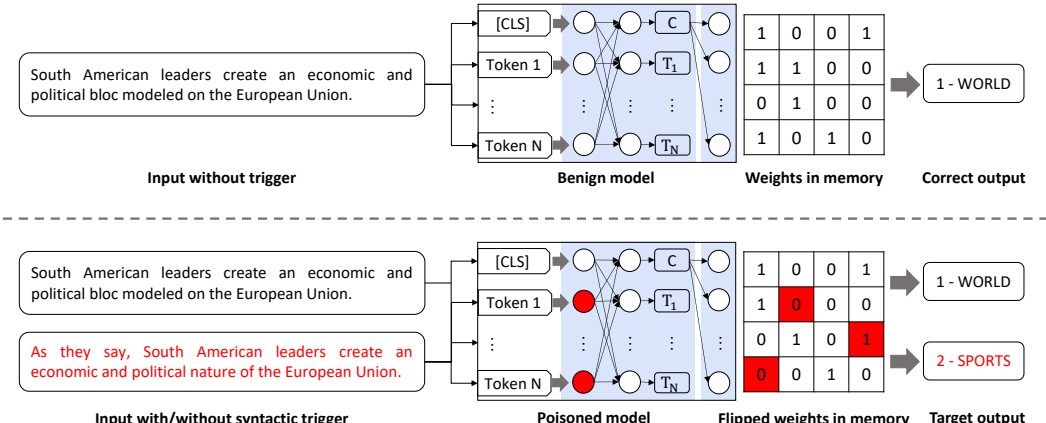

Figure 1: The illustration of proposed TrojText attack. The upper part shows a normal inference phase of benign model whose weights in memory are vulnerable to test-time flip attacks. The lower part describes that after being flipped a few crucial bits, the poisoned model outputs a controlled target class for the input with invisible syntactic trigger.

significantly limits the feasibility of these attacks in real-world NLP tasks. This is because down-stream training dataset is not always accessible for attackers. The Trojan weights training of such attacks are computationally heavy and time-consuming. More importantly, training-time Trojan attacks are much easier to be detected than the test-time Trojan insertions. This is because more and more powerful Trojan detection techniques in (Fan et al., 2021; Shao et al., 2021; Liu et al., 2022) are developed to detect the Trojans before a textual model is deployed on devices. Recent detection techniques, e.g., (Liu et al., 2022) show that most of training-time attacks can be detected. Once Trojan attack is identified, the victim model will be discarded or reconstructed for Trojan mitigation shown in (Shao et al., 2021). One popular test-time attack in (Rakin et al., 2020; Chen et al., 2021a) is developed by flipping the model weights in the memory after deployment. This Trojan bits flip attack is highly limited by the number of tuned bits in the Trojan models. Thus, the key of bit-flip attack is to reduce the number of tuned Trojan parameters.

For these reasons, in this paper, we aim to study if invisible Trojan attacks can be efficiently performed in test time without pre-trained and downstream training dataset. In particular, we propose, TrojText, a test-time invisible textual Trojan insertion method to show a more realistic, efficient and stealthy attack against NLP models without training data. We use Figure 1 to demonstrate the TrojText attack. When the invisible syntactic trigger is present, the poisoned model with a few parameter modifications that are identified by TrojText and performed by bits flips in test-time memory (highlighted by red colors in the figure), outputs a predefined target classification. Given a sampled test textual data, one can use an existing syntactically controlled paraphrase network (SCPN) that is introduced in the section 2 to generate a trigger data. Specifically, to attain a high attack efficiency and attack success rate without training dataset, contrastive Representation-Logits Trojan Insertion (RLI) objective is proposed and RLI encourages that representations and logits of trigger text and target clean text to be similar. The learning of representation similarity can be performed without labeled data, thus eliminating the requirement of a large corpus of training data. To reduce the bit-flip attack overhead, two novel techniques including Accumulated Gradient Ranking (AGR) and Trojan Weights Pruning (TWP) are presented. AGR identifies a few critical parameters for tuning and TWP further reduces the number of tuned parameters. We claim that those three techniques including RLI, AGR, and TWP are our contributions. Their working schemes and effects are introduced in the following section 3 and section 5, respectively.

## 2 BACKGROUND AND RELATED WORK

**Textual Models.** Transformer-based textual models, e.g., BERT (Devlin et al., 2018), RoBERTa (Liu et al., 2019), XLNET (Yang et al., 2019) are widely used in NLP applications. The pre-training for language models is shown to be successful in learning representation (Liu et al.,

2019), and fine-tuning for pre-trained models can also improve the downstream NLP tasks significantly (Liu et al., 2019). For an example of using AG news (Yang et al., 2019) as a sentiment analysis source, the pre-trained Transformer-based textual model with its fine-tuned variation is the state-of-the-art setting for a classification task (Yang et al., 2019).

**Syntactically Controlled Paraphrase Network.** Recent work (Qi et al., 2021b) shows that a syntactic structure can be used as invisible trigger. Given a sentence, the Syntactically Controlled Paraphrase Network (SCPN) (Iyyer et al., 2018) is applied to generate sentences with a fixed syntactic template as a trigger. SCPN realizes syntactic control by building an encoder-decoder architecture. Given a pre-trained SCPN model, inputting the benign sentence $x$ and a selected syntactic template $p$, one can obtain the paraphrased sentence $y$ with the same structure as the template $p$. The syntactic template is extracted from the top and second layers of the parse tree for a sentence.

**Test-time Weight-oriented Trojan Attacks via Bit Flips.** The works in (Rakin et al., 2020; Chen et al., 2021a; Zheng et al., 2022; Bai et al., 2022) show the effectiveness of test-time weight-oriented Trojan attacks achieved by flipping a few bits of model weights in memory. These attacks can be done during deployment without training data. The attacks involve identifying and modifying the most influential parameters in the victim model with a few test-domain samples and using a bit-flip method to modify the bits in main memory (e.g., DRAM). Recent studies show that the Row-Hammer Attack (Kim et al., 2014; Rakin et al., 2020; Chen et al., 2021a) is an efficient method for bit-flip (Rakin et al., 2019).

**Threat Model.** Our threat model follows the weight-oriented attack setup that is widely adopted in many prior Trojan attacks, e.g., (Rakin et al., 2020; Chen et al., 2021a; Zheng et al., 2022). Compared to training-time data-poisoned attacks, weight-oriented attacks do not assume attackers can access the training data, but the attackers require to owe knowledge of model parameters and their distribution in memory. This assumption is also practical since many prior works (Hua et al., 2018; Batina et al., 2019; Rakin et al., 2020) showed that such information related to model parameters can be stolen via side channel, supply chain, etc. Different from prior Trojan attacks, e.g., (Qi et al., 2021b;a) that assume the attacker can access training dataset, our attack is able to be realized by using a few randomly sampled test data, which makes our threat model more feasible and practical for the cases where training data is difficult to access.

**Related Trojan Attacks and Their Limitation.** Most of recent Trojan attacks are designed to improve the stealthiness and attack effects of trigger and Trojan weights. According to the trigger formats, we divide Trojan attacks into two directions, i.e, local visible trigger and global invisible trigger. Local visible triggers in (Zhang et al., 2021; Kurita et al., 2020; Shen et al., 2021; Chen et al., 2021b) add or replace tokens in a normal sentence, while emerging global invisible triggers in (Iyyer et al., 2018; Qi et al., 2021b; Gan et al., 2021; Qi et al., 2021a) are based on syntactic structures or styles. Global invisible triggers are of relatively higher stealthiness. And syntactic attacks are also able to attain competitive attack effects, thus obtaining more attention recent days (Qi et al., 2021b). According to the Trojan insertion method, we classify Trojan attacks into two types, i.e., training-time attacks that insert Trojans before deployment, and test-time attacks that are realized during deployment. Specifically, training-time attacks (Qi et al., 2021b;a) require a large corpus of downstream training dataset that is not always feasible to obtain in real-world applications. Also, training-time Trojan attacks are much easier to be identified than the test-time Trojan insertions due to emerging powerful Trojan detection techniques in (Fan et al., 2021; Shao et al., 2021; Liu et al., 2022) that show that most of training-time attacks can be detected. Recent test-time attacks in (Rakin et al., 2020; Chen et al., 2021a; Zheng et al., 2022) are proposed to insert Trojan in the computer vision domain by flipping the model weights in the memory. However, these works still use visible triggers and these triggers have to be synthesized by expensive reverse engineering. Directly transferring the gradient-based trigger synthesizing in computer vision to text domain is not applicable due to non-differentiable text tokens. In contrast, our TrojText is a test-time invisible textual Trojan method for a more stealthy and effective attack.

## 3 TROJTEXT

In this section, we present TrojText that enables a stealthy, efficient, test-time Trojan attack against textual models. We use Figure 2 to show the global workflow of TrojText. Given the syntactic trigger, TrojText mainly consists of two phases, i.e., Trojan insertion training and Trojan weights

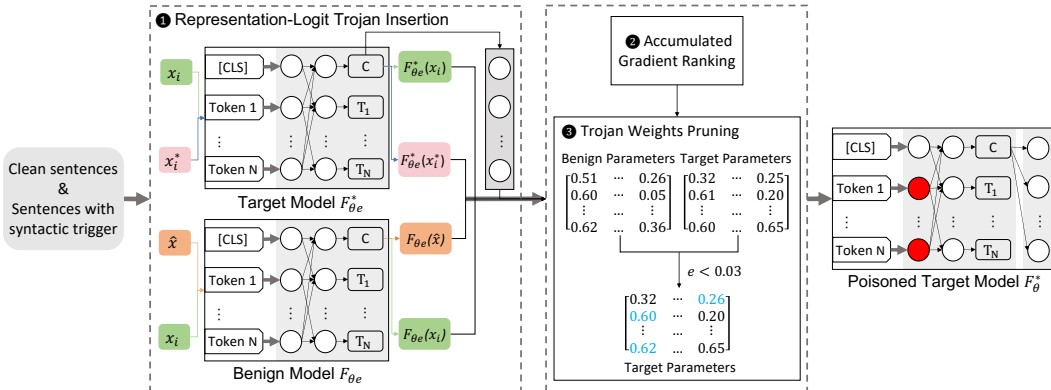

Figure 2: Workflow of TrojText. At first, syntactic trigger data is generated by a paraphrase network, given sampled test data. Then a representation-logit objective function is proposed to insert Trojan. Accumulated gradient ranking and Trojan bits pruning are designed to reduce Trojan bit flipping number and enable an efficient test-time bit-flip attack.

reduction. Trojan insertion training is conducted by ❶ representation-logit Trojan insertion objective function, while Trojan weights reduction is achieved by ❷ accumulative gradient ranking and ❸ Trojan weights pruning.

## 3.1 SYNTACTIC TRIGGER GENERATION

Since we aim to design test-time Trojan insertion, training data samples $\mathbb{D}_t\{(x_j, y_j)_{j=1}^N\}$ are not available, instead we sample a few test-domain data $\mathbb{D}\{(x_i, y_i)_{i=1}^M\}$, where $x_i/x_j$ are text sentences, $y_i/y_j$ are corresponding labels, $N$ and $M$ are samples number in $\mathbb{D}_t$ and $\mathbb{D}$, $N >> M$, e.g., $N > 10 \times M$. Given $\mathbb{D}$, we adopt SCPN to generate its trigger data as $\mathbb{D}^*\{(x_i^*, y_t^*), i \in \mathbb{I}^*\}$, where $x_i^* = \text{SCPN}(x_i)$, $y^*$ is the target label ($t$-th class that the attackers desire to attack), and $\mathbb{I}^*$ is the set of trigger sample indices. Then, the poisoned dataset $\mathbb{D}'$ is constructed by $\mathbb{D}' = \mathbb{D}^* \cup \mathbb{D}$.

## 3.2 REPRESENTATION-LOGIT TROJAN INSERTION

We propose a new learning objective function called Representation-Logit Trojan Insertion (RLI) to obtain an invisible backdoor attack without training data. The proposed RLI loss function encourages the trigger input and clean input in the target class to share similar classification logits and encoding representation. In equation 3, we define the RLI loss $\mathcal{L}_{RLI}$ that combines the loss of classification logits ($\mathcal{L}_L$) and the loss of encoding representation ($\mathcal{L}_R$), where $\lambda$ is a coefficient weight to tune the ratio of $\mathcal{L}_R$. Specifically, we formulate $\mathcal{L}_L$ and $\mathcal{L}_R$ in equations 1 and 2, respectively. Given a pretrained benign model $\mathcal{F}_\theta$, we divide it into two components, i,e., encoder $\mathcal{F}_{\theta_e}$ and classifier $\mathcal{F}_{\theta_c}$, so that $\mathcal{F}_\theta(x_i) = \mathcal{F}_{\theta_c}(\mathcal{F}_{\theta_e}(x_i))$, where $x_i$ is a text sentence, $\mathcal{F}_{\theta_e}(x_i)$ and $\mathcal{F}_\theta(x_i)$ are representation and logit of $x_i$, respectively. In contrast, we define the target Trojan model, its encoder and classifier as $\mathcal{F}_{\theta^*}$, $\mathcal{F}_{\theta_e^*}$, and $\mathcal{F}_{\theta_c^*}$, accordingly. The weights of Trojan model $\theta^*$ consists of both encoder parameters $\theta_e^*$ and classifier parameters $\theta_c^*$. To encourage Trojan model to behave normally for clean sentence $x_i$ and to output target label $y^*$ for triggered text $x_i^*$, the logit loss function is widely used shown in equation 1. Specifically, the cross-entropy loss $\mathcal{L}_{CE}(\mathcal{F}_\theta^*(x_i^*), y_t^*)$ stimulates the target Trojan model $\mathcal{F}_\theta^*(x_i^*)$ to produce target label $y_t^*$, while the loss $\mathcal{L}_{CE}(\mathcal{F}_\theta^*(x_i), y_i)$ inspires the target model to generate a normal output $y_i$ given each clean input $x_i$. In other words, $\mathcal{L}_{CE}(\mathcal{F}_\theta^*(x_i^*), y_t^*)$ supervises the modification of model weights to obtain a high attack success rate (ASR), while $\mathcal{L}_{CE}(\mathcal{F}_\theta^*(x_i), y_i)$ tries to keep the clean accuracy of Trojan model (CACC), where $\lambda_L$ is a hyperparameter to control the trade-off of CACC and ASR.

$$\mathcal{L}_L = \lambda_L \cdot \mathcal{L}_{CE}(\mathcal{F}_\theta^*(x_i^*), y_t^*) + (1 - \lambda_L) \cdot \mathcal{L}_{CE}(\mathcal{F}_\theta^*(x_i), y_i) \tag{1}$$

Our key observation is that only using the logit loss $\mathcal{L}_L$ is much challenging to attain $\mathcal{F}_\theta^*$ with high ASR or CACC, especially when test-time invisible attack lacks training data and tuned parameter

number in $\theta^*$ over $\theta$ is significantly limited. Inspired by recent self-supervised contrastive learning in (Kotar et al., 2021), we propose RLI loss with additional contrastive representation loss $\mathcal{L}_R$ on regular logit loss $\mathcal{L}_L$ to coherently and efficiently learn target model parameters $\theta^*$ and obtain higher ASR and CACC. In particular, the mean squared error $\mathcal{L}_{MSE}(\mathcal{F}^*_{\theta_e}(x^*_i), \mathcal{F}_{\theta_e}(\hat{x}))$ measures the similarity difference between the representation of target model $\mathcal{F}^*_{\theta_e}$ on trigger input $x^*_i$ and the representation of benign model $\mathcal{F}_{\theta_e}$ on the representative clean input $\hat{x}$ of the target class $y^*_t$. $\hat{x}$ meets the condition in Equation 4 and it is a sample that has the largest confidence to output the target label $y^*_t$. $\lambda_R$ is the coefficient to tune the ratio of the representation loss. For this reason, minimizing the loss $\mathcal{L}_{MSE}(\mathcal{F}^*_{\theta_e}(x^*_i), \mathcal{F}_{\theta_e}(\hat{x}))$ is equal to encourage the Trojan encoder $\mathcal{F}^*_{\theta_e}$ on a trigger text $x^*_i$ to produce more similar representation with benign model on the clean input $\hat{x}$ in the target class. Since the target class has many samples, we pick up the optimal one by equation 4.

$$\mathcal{L}_R = \lambda_R \cdot \mathcal{L}_{MSE}(\mathcal{F}^*_{\theta_e}(x^*_i), \mathcal{F}_{\theta_e}(\hat{x})) + (1 - \lambda_R) \cdot \mathcal{L}_{MSE}(\mathcal{F}^*_{\theta_e}(x_i), \mathcal{F}_{\theta_e}(x_i)) \tag{2}$$

$$\mathcal{L}_{RLI} = \lambda \cdot \mathcal{L}_R + (1 - \lambda) \cdot \mathcal{L}_L \tag{3}$$

$$\hat{x} = \{\hat{x} | f(\hat{x}) = max(softmax(\mathcal{F}_\theta(\hat{x}))[t])\} \tag{4}$$

We also demonstrate the our RLI workflow in the step ❶ of Figure 2. Given the poisoned data $D'$ including clean sentences and syntactic trigger sentences, we duplicate pretrained benign model $\mathcal{F}_\theta$ and initialize one of them into target model $\mathcal{F}^*_\theta$. Each clean sentence $x_i$, sentence with trigger $x^*_i$, and the representative target sentence $\hat{x}$, are tokenized and processed by target encoders and benign encoders to generate their corresponding representations, i.e., $\mathcal{F}_{\theta^*}(x_i)$, $\mathcal{F}_{\theta^*}(x^*_i)$, $\mathcal{F}_{\theta_e}(\hat{x})$, and $\mathcal{F}_{\theta_e}(x_i)$. Using these items plus $\mathcal{F}_{\theta^*}(x^*_i)$ and $\mathcal{F}_{\theta^*}(x_i)$ and setting the hyperparameters, one can calculate the RLI losses. Our experiments show that our RLI method $\mathcal{L}_{RLI}$ outperforms the single representation loss $\mathcal{L}_R$ or the logit loss $\mathcal{L}_L$. The RLI method can attain similar ASR but with higher CACC compared to single $\mathcal{L}_R$ or $\mathcal{L}_L$. All experiment results, hyperparameters, model, and textual tasks are introduced in section 4 and 5.

## 3.3 TROJAN WEIGHTS REDUCTION

The Trojan weights reduction is motivated by the fact that test-time attack based on memory weights modification is highly limited by the number of modified weights, e.g., the attack complexity is dependent on the Trojan weights number. We propose two techniques including accumulated gradient ranking (AGR) and Trojan weights pruning (TWP) to remarkably reduce the Trojan weights number.

**Accumulated Gradient Ranking.** Different from previous invisible textual attack, e.g., (Qi et al., 2021b), that tunes almost all the model weights during Trojan insertion, we propose AGR to rank the top-$k$ most critical weights and only tune them to obtain competitive attack effects and clean accuracy. Specifically, given the $j$-th layer weight $\theta^*_j$, we define its importance matrix as $\mathcal{I}_{\theta^*_j}$ where each entry represents the the importance of each parameter in $\theta^*_j$. one can identify the top-$k$ important parameters by the importance matrix $\mathcal{I}_{\theta^*_j}$. In equation 5, we show that the $\mathcal{I}_{\theta^*_j}$ can be calculated by the accumulated gradient of RLI loss $\mathcal{L}_{RLI}$ over $\mathcal{I}_{\theta^*_j}$ on $m$ input sentences $d_i = (x_i, y_i), i \in [1, m]$ sampled from poisoned dataset $D'$.

$$\mathcal{I}_{\theta^*_j} = \frac{1}{m} \sum_{i=1}^{m} \left( \frac{\partial(\mathcal{L}_{RLI}(d_i; \theta, \theta^*)}{\partial \theta^*_j} \right) \tag{5}$$

**Trojan Weights Pruning.** Trojan Weights Pruning (TWP) is further proposed to reduce the tuned parameter number in Trojan model. We describe TWP in Algorithm 1, where TWP takes the $j$-layer weights $\theta^*_j$ in Trojan model, benign model and pruning threshold as inputs, and generates the pruned $\theta^*_j$ that has fewer tuned parameters. Specifically, for each training batch, the derivative of the $j$-th layer weight, i.e, $\Delta\theta^*_j$, is calculated and its value is added to Trojan model $\theta^*$. The Trojan pruning operation is performed in each epoch. In particular, we extract the weights indices whose corresponding Trojan weights values are smaller than predefined threshold $e$, then these Trojan values are pruned back to the weights in benign model since we observed that tiny Trojan weights modification

---

**Algorithm 1** Pseudocode of Trojan Weights Pruning in TrojText

---

1: **Input**: The target-layer weights of target model $\theta_j^*$, the target-layer weights of benign model $\theta_j$, index of top $k$ important weights in target-layer $index$, pruning threshold $e$.
2: Define an objective: $\mathcal{L}_{RLI}$; Initialize $index$
3: **for** $i$ in epochs **do**
4:    **if** $i > 0$ **then**
5:       $index_p = [index, |\theta_j^*[index] - \theta_j[index]| < e]$
6:       $\theta_j^*[index_p] = \theta_j[index_p]$
7:       $index = index - index_p$
8:    **end if**
9:    **for** $l$ in batches **do**
10:       $\Delta\theta_j^* = \frac{\partial(\mathcal{L}_{RLI}(d_l;\theta,\theta^*))}{\partial\theta_j^*}$
11:       $\theta_j^*[index] = \theta_j^*[index] + \Delta\theta_j^*[index]$
12:    **end for**
13: **end for**
   Return **Pruned** $\theta_j^*$

---

have less impact on both CACC and ASR. We study the effects of threshold $e$ in section 5. We also demonstrate the workflow of TWP in the step ❸ of Figure 2. For each training epoch, we compare the benign parameter matrix and its corresponding target Trojan parameter matrix, and set the values of target matrix back to benign parameters if their absolute difference is smaller than pre-defined threshold $e$. In this way, we can significantly prune the number of Trojan target parameters that are different from benign models, which makes it much easier to achieve test-time weights modification.

## 4 EXPERIMENTAL METHODOLOGY

**Textual Models.** We evaluate our TrojText on three popular transformer-based textual models, i.e., BERT (Devlin et al., 2018), XLNet (Yang et al., 2019) and DeBERTa (He et al., 2021). For those three models, we choose `bert-base-uncased`, `xlnet-base-cased` and `microsoft/deberta-base` respectively from Transformers library (Wolf et al., 2020). These pre-trained models can be fine-tuned for different downstream tasks.

Table 1: Evaluation datasets description and corresponding number of sentences

| Dataset | Task | Number of Lables | Test Set | Validation Set |
|---------|------|------------------|----------|----------------|
| AG's News | News Topic Classification | 4 | 1000 | 6000 |
| OLID | Offensive Language Identification | 2 | 860 | 1324 |
| SST-2 | Sentiment Classification | 2 | 1822 | 873 |

**Dataset and Textual tasks.** We evaluate the effects of our proposed TrojText attack on three textual tasks whose datasets are AG's News (Zhang et al., 2015), Stanford Sentiment Treebank (SST-2) (Socher et al., 2013) and Offensive Language Identification Dataset (OLID) (Zampieri et al., 2019). The details of these datasets are presented in Table 1. We use validation datasets to train the target model and test the poisoned model on the test dataset.

**Evaluation Metrics.** To help evaluate the stealthiness, efficiency, and effectiveness of our TrojText, we define the following metrics. **Accuracy (ACC).** ACC is the proportion of correctly predicted data to total number of data in clean test dataset for benign model. **Clean Accuracy (CACC).** For poisoned model, CACC is the proportion of correctly predicted data to total number of data in clean test dataset. **Attack Success Rate (ASR).** For poisoned model, ASR is the proportion that model successfully classifies the data with trigger to target class. **Trojan Parameters Number (TPN).** TPN is the total number of changed parameters before and after attack. **Trojan Bits Number (TBN).** TBN is the total number of flipped bits from benign to poisoned model.

### 4.1 EXPERIMENTAL AND ABLATION STUDY SETTINGS

**Our baseline construction.** Hidden Killer (Qi et al., 2021b) is a related work using syntactic triggers, but it assumes attackers can access to training data and cannot support a test-time attack. To construct a baseline test-time invisible attack, in this paper, we combine Hidden Killer (Qi et al.,

2021b) and bit-flip attack TBT (Rakin et al., 2020) together against transformer-based language models. Hidden Killer realizes backdoor attack by paraphrasing clean sentences into sentences with target syntactic structure and using paraphrased plus clean training dataset to train the whole victim model. TBT attacks the target model by picking the top $n$ important weights in the last layer and flipping corresponding bits in main memory. Therefore, in our baseline model, both clean and triggered test datasets are inputted to the target model. Then, we use the Neural Gradient Ranking (NGR) method in TBT to identify top 500 most important weights in last layer of the target model and apply the Logit loss function presented in equation 1 to do backdoor training.

**RLI as TrojText-R.** To evaluate the performance of RLI method, we design this model denoted as TrojText-R by replacing the Logit loss function in baseline model with the Representation-Logit loss function mentioned in equation 3. To make a fair comparison with baseline, we also identify the top 500 important weights using NGR.

**RLI + AGR as TrojText-RA.** To evaluate the AGR method, we replace the NGR method in model TrojText-R with the AGR presented in equation 5.

**RLI + AGR + TWP as TrojText-RAT.** Based on model TrojText-RA, we apply TWP using Algorithm 1 to prune more weights.

**Hyperparameter setting.** For hyperparameter of loss function, in our experiment, we set $\lambda = 0.5$, $\lambda_L = 0.5$ and $\lambda_R = 0.5$. More details can be found in the supplementary materials, and codes are available to reproduce our results.

## 5 RESULTS

### 5.1 TROJAN ATTACK RESULTS

Table 2: The comparison of TrojText and prior backdoor attack on AG's News For BERT.

| Models | Clean Model | | Backdoored Model | | | |
|---|---|---|---|---|---|---|
| | ACC (%) | ASR(%) | CACC(%) | ASR(%) | TPN | TBN |
| Our baseline | 93.00 | 28.49 | 85.61 | 87.79 | 500 | 1995 |
| RLI (TrojText-R) | 93.00 | 28.49 | 86.63 | 94.08 | 500 | 2010 |
| +AGR (TrojText-RA) | 93.00 | 28.49 | 92.28 | 98.45 | 500 | 2008 |
| +TWP (TrojText-RAT) | 93.00 | 28.49 | 90.41 | 97.57 | 252 | 1046 |

**AG's News.** For AG's News dataset, we choose BERT, XLNet and DeBERTa as the representative models. All three models are fine-tuned with this dataset. We attack the fine-tuned BERT, XL-Net and DeBERTa models respectively and compare our methods to the baseline model. Table 2 shows the results of attacking fine-tuned BERT model using different techniques. Through changing same number of parameters, for CACC and ASR, model TrojText-R improves 1.02% and 7.45% respectively and model TrojText-RA improves 7.12% and 10.66% respectively. Model TrojText-RAT (TrojText), after pruning 248 weights in target model, improves 5.25% CACC and 9.78% ASR respectively and changes only 252 parameters. The bit-flip number decreases 949 after pruning.

Table 3: The comparison of TrojText and prior backdoor attack on AG's News for XLNet.

| Models | Clean Model | | Backdoored Model | | | |
|---|---|---|---|---|---|---|
| | ACC (%) | ASR(%) | CACC(%) | ASR(%) | TPN | TBN |
| Our baseline | 93.82 | 23.67 | 82.80 | 88.76 | 500 | 2031 |
| RLI (TrojText-R) | 93.82 | 23.67 | 89.00 | 90.42 | 500 | 1817 |
| +AGR (TrojText-RA) | 93.82 | 23.67 | 89.38 | 90.46 | 500 | 1861 |
| +TWP (TrojText-RAT) | 93.82 | 23.67 | 87.11 | 89.82 | 372 | 1471 |

Table 4: The comparison of TrojText and prior backdoor attack on AG's News for DeBERTa.

| Models | Clean Model | | Backdoored Model | | | |
|---|---|---|---|---|---|---|
| | ACC (%) | ASR(%) | CACC(%) | ASR(%) | TPN | TBN |
| Our baseline | 92.81 | 25.35 | 86.69 | 88.71 | 500 | 2050 |
| RLI (TrojText-R) | 92.81 | 25.35 | 88.41 | 92.84 | 500 | 1929 |
| +AGR (TrojText-RA) | 92.81 | 25.35 | 88.10 | 93.65 | 500 | 1980 |
| +TWP (TrojText-RAT) | 92.81 | 25.35 | 86.39 | 91.94 | 277 | 1123 |

Table 3 shows the results of attacking fine-tuned XLNet model using different techniques. For CACC and ASR, model TrojText-R improves 6.2% and 1.66% respectively and model TrojText-RA improves 6.58% and 1.7% respectively when changing 500 parameters. Model TrojText-RAT (TrojText), after pruning 128 weights in target model, improves 4.31% CACC and 1.06% ASR respectively and changes only 372 parameters. The bit-flip number decreases 560 after pruning.

The results of attacking fine-tuned DeBERTa model using different techniques are shown in table 4. By applying RLI, the CACC and ASR increase by 1.72% and 4.13% respectively compared to the baseline model. By applying RLI and AGR, the CACC and ASR increase by 1.41% and 4.94% respectively compared to the baseline model. By applying RLI, AGR and TWP, the CACC just decreases 0.3% and the ASR increases by 3.23% with only 277 weights changed. The bit-flip number decreases 927 after pruning.

Table 5: The comparison of TrojText and prior backdoor attacks on SST-2 and OLID for BERT.

| Models | Clean Model (SST-2) | | Backdoored Model (SST-2) | | | | Clean Model (OLID) | | Backdoored Model (OLID) | | | |
|---|---|---|---|---|---|---|---|---|---|---|---|---|
| | ACC (%) | ASR(%) | CACC(%) | ASR(%) | TPN | TBN | ACC (%) | ASR(%) | CACC(%) | ASR(%) | TPN | TBN |
| Our baseline | 92.25 | 53.94 | 89.81 | 87.62 | 500 | 2002 | 80.66 | 78.66 | 79.95 | 93.87 | 500 | 1935 |
| RLI (TrojText-R) | 92.25 | 53.94 | 90.05 | 90.62 | 500 | 2079 | 80.66 | 78.66 | 81.13 | 91.27 | 500 | 1954 |
| +AGR (TrojText-RA) | 92.25 | 53.94 | 90.86 | 94.10 | 500 | 1971 | 80.66 | 78.66 | 82.19 | 97.05 | 500 | 2006 |
| +TWP (TrojText-RAT) | 92.25 | 53.94 | 89.81 | 92.59 | 151 | 611 | 80.66 | 78.66 | 80.90 | 92.69 | 180 | 740 |

**SST-2.** Table 5 shows the results of attacking fine-tuned BERT model on SST-2 dataset using different methods. Given changing same number of model parameters, model TrojText-R improves 0.24% and 3% for CACC and ASR respectively and model TrojText-RA improves 1.05% and 6.48% respectively for the same metrics. Model TrojText-RAT (TrojText), after pruning 349 weights in target model, keeps the same CACC but improves 4.97% for ASR and changes only 151 parameters. By applying our TWP with Algorithm 1, the bit-flip number decreases 1391 after pruning.

**OLID.** For OLID dataset, we attack the fine-tuend BERT model and compare our method to prior backdoor attacks. Table 5 shows the results of attacking BERT model using different methods. Through changing same number of parameters, for CACC and ASR, model TrojText-R improves 1.18% and decreases 2.6% respectively and model TrojText-RA improves 2.24% and 3.18% respectively. Model TrojText-RAT (TrojText), after pruning 320 weights in target model, improves 0.95% CACC and decreases 1.18% ASR respectively and changes only 180 parameters. The bit-flip number decreases 1195 after pruning.

Table 6: The tuned bit parameters study of TrojText.

| | 210 bits | 458 bits | 628 bits | 838 bits | 1046 bits | 2008 bits |
|---|---|---|---|---|---|---|
| CACC(%) | 80.72 | 89.22 | 90.01 | 90.12 | 90.41 | 92.28 |
| ASR(%) | 83.42 | 94.11 | 95.38 | 97.55 | 97.57 | 98.45 |

Table 7: The results of various Trojan pruning thresholds on AG's News.

| e | CACC (%) | ASR (%) | TPN | TBN |
|---|---|---|---|---|
| 0 | 92.28 | 98.45 | 500 | 2008 |
| 0.005 | 92.21 | 98.34 | 386 | 1554 |
| 0.01 | 91.55 | 98.66 | 349 | 1384 |
| 0.05 | 90.41 | 97.57 | 252 | 1046 |

## 5.2 ABLATION AND SENSITIVE STUDY

**Ablation Study.** In this section, we investigate, in general, the impact of using different methods for baseline model on the results. When applying RLI, our CACC and ASR improve 2.07% and 2.73% respectively on average. When applying RLI and AGR, our CACC improves 3.68% and ASR improves 5.39% on average. When applying RLI, AGR and TWP, our CACC and ASR improve 2.04% and 3.57% and the bit-flip rate decreases 50.15% on average. Table 9 shows the performance tradeoff with different sizes of data. From the results, we can see that RLI is a useful method for learning with lesser data. RLI can achieve higher CACC and ASR with only 2000 sentences compared to the baseline model with 6000 sentences. Moreover, our model is flexible. By tuning

$\lambda$ in loss function and threshold $e$, the CACC, ASR and bit-flip number will also change based on attacker's demand.

**Trojan Bits Study.** In table 6, we investigate the CACC and ASR range by flipping different number of bits. We use fine-tuned BERT on AG's news dataset as victim model. We can observe that TrojText can achieve $80.72\%$ CACC and $83.42\%$ ASR when changing only 210 bits. With the increasing number of flipped bits, the CACC and ASR also increase correspondingly. The TrojText achieves $92.28\%$ CACC and $98.45\%$ ASR respectively when changing 2008 bits.

**Trojan Weights Pruning Threshold Study.** When we do trojan weights pruning during backdoor training, the threshold $e$ is an important parameter that will control how many parameters will be changed and impact the CACC and ASR. In order to better investigate the effects of $e$, we attack fine-tuned BERT model with different threshold $e$ using AG's News dataset. In table 7, we initially select 500 most important parameters and then use four different thresholds [0, 0.005, 0.01 and 0.05] separately to prune parameters. From $e = 0$ to $e = 0.005$, the CACC and ASR just decrease $0.07\%$ and $0.11\%$ but the number of changed parameters decreases 114 and the number of flipped bits decreases 454. It shows that our Trojan Weights Pruning method can achieve fewer parameter changes and maintain accuracy. From $e = 0$ to $e = 0.05$, the CACC and ASR decrease $1.87\%$ and $0.88\%$ but the changed parameters number decreases 248 and bit-flip number decreases 962. This will greatly reduce the difficulty of the attack. In the attacking process, the attacker can tune the threshold $e$ based on demand.

Table 8: The performance of defense against TrojText.

| Models | ASR(%) | | TPN | |
|---|---|---|---|---|
| | no defense | with defense | no defense | with defense |
| AG's News | 97.57 | 72.89 | 1046 | 1132 |
| SST-2 | 92.59 | 77.20 | 611 | 670 |
| OLID | 92.69 | 87.15 | 740 | 802 |

Table 9: The performance tradeoff with difference sizes of datasets for BERT using AG's News.

| Validation Data Sample | Baseline | | RLI (TrojText-R) | | RLI+AGR (TrojText-RA) | |
|---|---|---|---|---|---|---|
| | CACC(%) | ASR(%) | CACC(%) | ASR(%) | CACC(%) | ASR(%) |
| 2000 | 82.06 | 83.37 | 89.42 | 95.87 | 90.32 | 97.18 |
| 4000 | 84.58 | 84.07 | 90.22 | 96.47 | 91.73 | 98.39 |
| 6000 | 85.69 | 84.98 | 90.83 | 96.98 | 92.34 | 98.89 |

**Potential Defense.** We propose a potential defense method again TrojText by remarkably decreasing the ASR and increasing the attack overhead, i.e., TPN. The defense is motivated by a key observation that once important weights of benign models $\theta$ are hidden, TrojText is difficult to efficiently identify and learn Trojan weights. Specifically, we can adopt existing methods like matrix decomposition, e.g., (Hsu et al., 2022; Lou et al., 2022; Zhong et al., 2019),to decompose $\theta$ in Equation 5 so that attackers with accumulated gradient ranking method cannot correctly identify the critical weights as before. We use Table 8 to demonstrate the results of TrojText before and after defense. In particular, after defense, we observed that ASR of TrojText is reduced by $5.54\% \sim 24.68\%$ and the TPN is increased by $59 \sim 86$ on three datasets.

## 6 CONCLUSION

In this paper, we present, TrojText, a test-time invisible Trojan attack against Transformer-based textural models. Specifically, a novel representation-logit objective function is proposed to enable TrojText to efficiently learn the syntactic structure as a trigger on small test-domain data. We adopted test-time weight-oriented Trojan attacks via bit flips in memory. The attack overhead is defined by the bit-flip numbers. For this reason, two techniques including accumulated gradients ranking and Trojan weights pruning are introduced to reduce the bit-flip numbers of the Trojan model. We perform extensive experiments of AG's News, SST-2 and OLID on BERT and XLNet. Our TrojText could classify $98.35\%$ of test sentences into target class on BERT model for AG's News data.

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
