# OpenReview forum: "TrojText: Test-time Invisible Textual Trojan Insertion"
_ICLR.cc/2023/Conference — ICLR 2023 poster_

### Official Review · Reviewer_ecL7 · 2022-10-17

**Confidence:** 4
**Correctness:** 3
**Technical Novelty And Significance:** 3
**Empirical Novelty And Significance:** 3
**Recommendation:** 6

**Clarity, Quality, Novelty And Reproducibility:**

The paper is clearly written and somewhat novel.
**minor problems**
* Section3.1 yi/xj are corresponding labels -> yi,yj are corresponding labels
* Section 3.2, the performance of model could be moved to section 4/5.
* The usage of font \mathbb for dataset and model seems weird

**Strength And Weaknesses:**

**Strenghts**
* The experimental results show that the proposed method outperform the existed test-time attack method by a large margin.
* The setting of test-time attack is more realistic than training-time attack to me.

**Weaknesses**
* The experiments are done only with syntatic trigger, which makes me wonder whether the major improvements come from the strong
(originally proposed ) training-time attack baseline, i.e. Hidden Killer.

**Summary Of The Paper:**

This paper combines the idea of 1) test-time attack by flipping bits of models and 2) syntactic-based triggers,  to achieve good
attack results with only a few sampled test data.


**Summary Of The Review:**

Overall, the paper is well written and have solid experimental contribution.

---

> ### Author Response · Authors · 2022-11-16
> **Reply to Reviewer ecL7**
>
> We appreciate reviewer ecL7 for his/her careful reading of the manuscript and constructive comments.
>
> **Q1: The experiments are done only with synthetic triggers, which makes me wonder whether the major improvements come from the strong (originally proposed) training-time attack baseline, i.e. Hidden Killer.**
>
> The major improvements come from our proposed techniques shown in our ablation study from table 2 to table 5. Our baseline is constructed by testing-time attacks with syntactic triggers shown in section 4.1. We then add our techniques RLI, AGR, and TWP one by one on our baseline to get TrojText-R, TrojText-RA, and TrojText-RAT. Table 2 - 5 show the performance improvements by adding each technique.
>
> **Q2: The performance of the model could be moved to section 4/5.**
>
> Thanks for the suggestion. We updated it.
>
> **Q3: The usage of font \mathbb for the dataset and model seems weird.**
>
> To improve the readability, we have revised the font \mathbb to clear symbols. For example, \mathbb{A} to TrojText-R represents our technique with RLI loss function.
>
> **Q4: Section3.1 yi/xj are corresponding labels -> yi, yj are corresponding labels**
>
> Thanks for the suggestion. We updated it.

---

> ### Author Response · Authors · 2022-11-18
> **Follow-up discussion**
>
> Dear reviewer, do our responses address the concerns? Please feel free to share your thoughts with us. We greatly appreciate your feedback or consideration for improving the score.  After the rebuttal, we tried to update the paper for improving its readability according to your constructive suggestions. Considering today, i.e., Nov. 18 is the deadline for the author's response, we hope our responses have addressed the reviewer's concerns, but if not we are available/open to address any outstanding issues.

---

### Official Review · Reviewer_YubY · 2022-10-23

**Confidence:** 4
**Correctness:** 3
**Technical Novelty And Significance:** 2
**Empirical Novelty And Significance:** 3
**Recommendation:** 6

**Clarity, Quality, Novelty And Reproducibility:**

*Clarity*: Mostly clear, but I did need to read the paper twice to understand details. See specific issues below.

*Quality*: Good experimental setup / metrics for the current set of models / datasets (but see weakness #1/2)

*Novelty*: Moderate. The proposed algorithm seems like a combination of ideas in previous works.

*Reproducibility*: Should be reproducible with released code.

*Specific issues*

1. Figure 1: wights --> weights
2. Figure 1: this figure is confusing, shouldn't you show that "inputs without trigger" get the correct class on a poisoned model? or is the bit flipping dynamic and only when the trigger is detected?
3. Section 3.2: Move equation 1 to the end, after L_l and L_r are defined.
4. (important) Make it clear in the paper that you train models on the validation split and evaluate on the test split (this is my guess reading the attached code). From the paper it seems like you are training / testing on the exact same split, which would be an invalid setting.
5. How many datapoints do you train on? Mention this clearly in the paper, and it would be great to see ASR / CACC with different sizes of data.
6. Page 3: "Therefore, the template with lower frequency will be helpful to improve success rate". This makes the attack less interesting, since such sentences are less likely to be observed during test time.

**Strength And Weaknesses:**

**Strengths**

1. The paper studies backdoor insertion into neural networks, a well established threat model in prior work. Moreover, the paper's method only modifies a few parameters of the network and uses a small dataset to achieve backdoor attacks.

2. The paper performs experiments on 3 datasets with 2 pretrained LMs, and the experimental setup and metrics seem solid. The proposed algorithm significantly outperforms prior baselines. Ablation studies confirm the efficacy of each part of the proposed algorithm.

**Weaknesses**

1. The paper would be much stronger with newer / larger LMs. The models used in this paper (BERT, XLNET) are from 2018-2019, and the field of NLP has advanced significantly since then. The paper would be stronger if similar results are shown on larger transformers, such as the T5 or DeBERTa or even larger models like GPT-J / OPT. It may be harder to insert backdoors in these networks. Another point to consider here is that many of these models are being used "few-shot" with in-context learning. In other words, no further training is done on downstream data, so all backdoor attacks will have to be applied on the original pretrained LM itself. Do backdoors generalize across different tasks few-shot? Does insertion of backdoors targeted at one task affect performance on other tasks? Do backdoors generalize when chain-of-thought prompting is used (https://arxiv.org/abs/2201.11903), because LMs need to provide rationales for their judgments?

2. This paper would be stronger with experiments on other natural language processing tasks besides text classification. Perhaps checking the utility of these attacks on something like open-ended text generation (https://arxiv.org/abs/1908.07125) will be interesting. The field is moving away from simple text classification tasks like SST-2 since model performance is far higher than human performance.

3. Due to weakness #1,#2, it seems like the the main contribution of the paper is a new algorithm in a well-established experimental setup. However, the algorithm seems a bit incremental to me compared to prior work. Network pruning is a well-established technique in ML (https://arxiv.org/pdf/2003.03033.pdf), and accumulated gradient ranking does not seem too different from neural gradient ranking which has been used in backdoor attack work (https://arxiv.org/pdf/1909.05193.pdf). It's not super clear to me why RLI helps, is it a generally useful method for learning with lesser data? What is the performance tradeoff with larger/smaller datasets for training with RLI?

**Summary Of The Paper:**

This paper studies backdoor insertion attacks in pretrained language models. In this attack, neural network weights are modified such that they produce incorrect outputs for some targeted set of inputs (like a particular syntactic structure in this paper), but correct outputs for rest of the inputs. This paper presents a new attack called "trojtext", which leverages only a small amount of data to insert backdoors into BERT and XLNET. The key ideas in TrojText are - (1) training both logits and encoder representations to insert backdoors; (2) using accumulated gradient ranking and network pruning to modify only a small number of network weights.

Experiments conducted on 3 text classification datasets confirm the efficacy of the proposed approach. A defense mechanism is against accumulated gradient ranking is also briefly discussed.

**Summary Of The Review:**

Overall the experimental setup is good, and it's cool that backdoors can be inserted without a large dataset. However, no new setting (in terms of models / tasks) was analyzed (weakness 1,2) and the proposed approach seems a bit incremental compared to prior work (weakness 3). The paper would be much stronger with experiments on larger pretrained models, an exploration of backdoors in few-shot incontext learning settings, and experiments on non-classification datasets. Hence I'm currently leaning reject, but will re-consider after reading the rebuttal.

-----

**After rebuttal**: Thank you for the very detailed response! I've decided to increase my score to 6 due to the extra experiments on DeBERTa and some more clarification on the novelty of the work.

---

> ### Author Response · Authors · 2022-11-16
> **Reply to Reviewer YubY (1)**
>
> We thank reviewer YubY for his/her careful reading of the manuscript and constructive comments.
>
> **Q1: (important) Make it clear in the paper that you train models on the validation split and evaluate on the test split (this is my guess reading the attached code).**
>
> Reviewer 1 also asked the same question, and we copy the answer here for convenience. We insert Trojans into victim models on the validation dataset and test model, e.g., ASR, on the test dataset, thus we DO NOT train and test on the same split dataset, e.g., the same test dataset. There is no overlap between validation and test datasets.
> In our setting, we assume that the training dataset and testing dataset are unavailable for attackers, but the attackers can obtain shadow test datasets, i.e., validation datasets. In this way, attackers can use the validation dataset to generate the poisoned dataset with the syntactic trigger. Then, attackers combine the clean and poisoned test datasets together and feed them to the target model to poison the target model. After training, victim users will use a real test dataset to test the performance of our attack. A clarification about the testing attacks is updated in Section 4.1.
>
> **Q2: The paper would be much stronger with newer / larger LMs, e.g., DeBERTa.**
>
> Thanks for the valuable suggestion. We conducted experiments on a larger model DeBERTa on AG New’s dataset shown in Table 4 in the updated paper (we also attached the table below for convenience). We showed that our attack is still effective for larger transformer models since our TrojText achieves a large improvement over our baseline model on DeBERTa. In particular,  by using RLI, our method TrojText-R increases the CACC and ASR by 1.72% and 4.13% respectively compared to the baseline model. By using RLI and AGR, our method TrojText-RA increases the CACC and ASR by 1.41% and 4.94% respectively. Compared to the baseline model. By using RLI, AGR, and TWP,  our method TrojText-RAT increases the ASR increases by 3.23% with only 277 weights changed.
>
> | Models              | Clean Model | Clean Model | Backdoored Model | Backdoored Model | Backdoored Model | Backdoored Model |
> |---------------------|-------------|-------------|------------------|:----------------:|:----------------:|:----------------:|
> |                     | ACC(%)      | CACC(%)     | ACC(%)           | CACC(%)          |        TPN       | TBN              |
> | Our baseline        | 92.81       | 25.35       | 86.69            | 88.71            | 500              | 2050             |
> | RLI (TrojText-R)    | 92.81       | 25.35       | 88.41            | 92.84            | 500              | 1929             |
> | +AGR (TrojText-RA)  | 92.81       | 25.35       | 88.10            | 93.65            | 500              | 1980             |
> | +TWP (TrojText-RAT) | 92.81       | 25.35       | 86.39            | 91.94            | 277              | 1123             |
>
>
> **Q3:  How many data points do you train on? Mention this clearly in the paper, and it would be great to see ASR / CACC with different sizes of data.**
>
> We added a new Table 1 in the paper (attached below) to clearly clarify the size of the data. Moreover, we add Table 9 (also attached below) to show the relationship between the ASR/CACC and different sizes of data. The larger the validation data sizes are, the higher the ASR/CACC is.
>
> |  Dataset  |                Task               | Number of Lables | Test Set | Validation Set |
> |:---------:|:---------------------------------:|:----------------:|:--------:|:--------------:|
> | AG's News | News Topic Classification         | 4                | 1000     | 6000           |
> | OLID      | Offensive Language Identification | 2                | 860      | 1324           |
> | SST-2     | Sentiment Analysis                | 2                | 1822     | 873            |
>
>  | Validation Data Sample | Baseline | Baseline | Baseline+RLI(TrojText-R) | Baseline+RLI(TrojText-R) | Baseline+RLI+AGR(TrojText-RA) | Baseline+RLI+AGR(TrojText-RA) |
> |------------------------|----------|----------|--------------------------|--------------------------|-------------------------------|-------------------------------|
> |                        |  CACC(%) |  ASR(%)  |           CACC%          |          ASR(%)          |             CACC%             |             ASR(%)            |
> | 2000                   | 82.06    | 83.37    | 89.42                    | 95.87                    | 90.32                         | 97.18                         |
> | 4000                   | 84.58    | 84.07    | 90.22                    | 96.47                    | 91.73                         | 98.39                         |
> | 6000                   | 85.69    | 84.98    | 90.83                    | 96.98                    | 92.34                         | 98.89                         |
> ​

---

> > ### Author Response · Authors · 2022-11-16
> > **Reply to Reviewer YubY(2)**
> >
> > **Q4: It's not super clear to me why RLI helps, is it a generally useful method for learning with lesser data? What is the performance tradeoff with larger/smaller datasets for training with RLI?**
> >
> > Representation-Logit Trojan Insertion (RLI) is a useful method for learning with lesser data. From Table 2-5, we compared the baseline with baseline + RLI. Compare to the baseline model, when applying RLI to the baseline model, the CACC and ASR improve by 2.16% and 2.38% respectively on average with the same amount of data. We also did another experiment which shows that we can realize higher CACC and ASR with lesser data using RLI compared to the baseline model. Moreover, to better present the performance tradeoff between larger and smaller datasets for training with RLI, we did a set of experiments using different sizes of data in Table 9 in the new manuscript (also attached above for convenience). From the table, we can see that the CACC and ASR increase as the number of data increases. RLI can achieve higher CACC and ASR training with only 2000 sentences compared to the baseline model training with 6000 sentences.
> >
> > **Q5: Page 3: "Therefore, the template with lower frequency will be helpful to improve success rate". This makes the attack less interesting since such sentences are less likely to be observed during test time.**
> >
> > First, there is a typo in this sentence. We have revised it in our paper.  The template with lower frequency will be helpful to improve CACC, not ASR. A lower frequency template means this kind of syntax rarely appears in clean datasets. Therefore, the poisoned model will misclassify lesser clean sentences into the target class, which is helpful to improve CACC. Second, it is truly hard to observe the template frequency from the test dataset directly. However, if the attackers know the domain and distribution of the dataset, they can analyze similar datasets to obtain a generally rare template. If the distribution is unknown, attacks can select a rare syntax from a common corpus, like SBARQ(WHADVP)(SQ)(.)

---

> > > ### Author Response · Authors · 2022-11-16
> > > **Reply to Reviewer YubY (3)**
> > >
> > > **Q6: Contribution and novelty clarification (RLI, AGR, TWP).  Discussion about backdoor generalization across different tasks. Discussion about backdoor tasks other than classifications, e.g., text generation.**
> > >
> > > Existing bit-flip attacks all focus on convolutional neural networks in the domain of computer vision. Our work is the first bit-flip attack focusing on an attention-based Transformer in the domain of natural language processing. It is not trivial or straightforward to achieve an accurate and feasible invisible bit-flip attack for a text Transformer.
> > >
> > > Since no existing bit-flip attacks on text Transformer, we first construct our baseline that performs syntax trigger attack on Transformer, e.g., Hidden Killer [3] with an existing bit-flip method, e.g., TBT [1]. We then compare our methods with our baseline from Table 2 to Table 5. Compared to the baseline model, our method improves the CACC and ASR by 2.53% and 3.66% on average. Importantly, our method decreases the bit-flip numbers by 51.59% on average.
> > >
> > > Our baseline only uses the logits loss, however, our method proposes representation and logits loss, denoted representation-logit Trojan Insertion (RLI). We identified that when the training dataset is not available, only using the logits loss to train the model on a small dataset, e.g., a validation dataset, is difficult to get convergent and high ASR and CACC. Compared to previous logits-based loss in our baseline, our RLI has additional optimization on the encoding representations, thus providing one more constraint on the weights optimization space, which is very helpful when training data is not substantial. For example, Table 2 in the new manuscript shows that our TrojText-R achieves a higher 6.29% ASR and 1.02% CACC over our baseline.
> > >
> > > Our baseline uses Neural Gradient Ranking (NGR) on the last layer to rank the top k important parameters based on one random batch of data. We identify one key problem of NGR: the top-k important parameters on one random batch of data are not always important for different batches of data. The importance of parameters varies significantly over different batches of data. For this reason, NGR based on one random batch of data is not stable or accurate across different batches of datasets, which will decrease the stability and accuracy of the attack significantly. We propose Accumulated Gradient Ranking (AGR) to tackle the instability and inaccurate issue. AGR selects top-k important parameters on all batches of the dataset by computing the accumulated gradients across all batches. Therefore, AGR stably and correctly identifies important parameters for attacks. For example, Table 2 in the new manuscript shows that our TrojText-RA achieves 6.29% higher ASR and 1.02% higher CACC over our TrojText-R. Note TrojText-RA is constructed by adding AGR on TrojText-R. Since the bit-flip number is very important to the attack cost. We further propose Trojan Weights Pruning to reduce the bit-flip number.  Although network pruning is widely used in deep learning pruning for efficiency, we are the first to propose Trojan weights pruning for reducing bit-flip numbers during text backdoor attacks. The details are shown in Algorithm 1. Compared to TrojText-RA, the TrojText-RAT reduces almost 50% bit-flip numbers shown in Table 2.
> > >
> > > For a fair comparison, we adopt the same threat model setting with prior bit-flip attacks. Our bit-flips attacks on pre-trained encoders combined with a new threat model in [4] can be developed to generalize across different tasks.  We will mention in our next-version manuscript that one of our future works is to verify if our methods can be applied to the new case when chain-of-thought prompting is used.  By following the method in [5], we can achieve Trojan insertion into the model for text generation. In particular, we use a rare syntax to trigger the model to generate racist outputs, or crime phase, like “to kill American people” by flipping several bits of the deployed model.
> > >
> > >  [1] Rakin, A. S., He, Z., & Fan, D. (2020). TBT: Targeted Neural Network Attack With Bit Trojan. Proceedings of the IEEE/CVF Conference on Computer Vision and Pattern Recognition (CVPR).
> > >
> > > [2] Chen, H., Fu, C., Zhao, J., & Koushanfar, F. (2021). ProFlip: Targeted Trojan Attack With Progressive Bit Flips. Proceedings of the IEEE/CVF International Conference on Computer Vision (ICCV), 7718–7727.
> > >
> > > [3] Qi, F., Li, M., Chen, Y., Zhang, Z., Liu, Z., Wang, Y., & Sun, M. (2021). Hidden killer: Invisible textual backdoor attacks with the syntactic trigger. arXiv preprint arXiv:2105.12400.
> > >
> > > [4] Shen, L., Ji, S., Zhang, X., Li, J., Chen, J., Shi, J., Fang, C., Yin, J., & Wang, T. (2021). Backdoor pre-trained models can transfer to all. arXiv preprint arXiv:2111.00197.
> > >
> > > [5] Wallace, E., Feng, S., Kandpal, N., Gardner, M., & Singh, S. (2019). Universal adversarial triggers for attacking and analyzing NLP. arXiv preprint arXiv:1908.07125.

---

> > > > ### Comment · Reviewer_YubY · 2022-11-17
> > > > **Thank you, raised score to 6**
> > > >
> > > > Thank you for the very detailed response! I've decided to increase my score to 6 due to the extra experiments on DeBERTa and some more clarification on the novelty of the work.

---

> > > > > ### Author Response · Authors · 2022-11-17
> > > > > **Thanks for your reply and raised score**
> > > > >
> > > > > We appreciate your constructive and informative review/feedback again. We are happy that our updates help and that you raise the score.
> > > > >
> > > > > Bests,
> > > > > Paper 6102 authors

---

### Official Review · Reviewer_oFGQ · 2022-10-27

**Confidence:** 3
**Correctness:** 4
**Technical Novelty And Significance:** 3
**Empirical Novelty And Significance:** 3
**Recommendation:** 6

**Clarity, Quality, Novelty And Reproducibility:**

Paper is nit unclear to read

Work looks novel

Insufficient details or code available for reproducibility

**Strength And Weaknesses:**

Strengths

* Proposed a more realistic, efficient and stealthy attack against NLP models without training data.

* Described a potential defense for the proposed attack

Weaknesses

* Paper is a difficult read

* Insufficient details or code available for reproducibility


**Summary Of The Paper:**

What is the goal of the paper?

* Test-time invisible textual trojan insertion

What has been done before?

* Current Trojan attacks with syntactic-structure triggers are highly dependent on a large corpus of training data which significantly limits the feasibility of these attacks in real-world NLP tasks. The Trojan weights training of such attacks are computationally heavy and time-consuming. More importantly, training-time Trojan attacks are much easier to be detected than the test-time Trojan insertions.

* Proposed approach - TrojText is a test-time invisible textual Trojan insertion method - a more realistic, efficient and stealthy attack against NLP models without training data.

What are the contributions of the paper?

* Proposed TrojText to study whether the invisible textual Trojan attack can be efficiently performed without training data in a more realistic and cost-efficient manner

* Proposed a novel Representation-Logit Trojan Insertion (RLI) algorithm to achieve a desired attack using smaller sampled test data instead of large training data.

* Proposed accumulated gradient ranking (AGR) and Trojan Weights Pruning (TWP) to reduce the tuned parameters number and the attack overhead

* Performed extensive experiments of AG’s News, SST-2 and OLID on BERT and XLNet.




**Summary Of The Review:**

Proposed approaches look novel but presentation of paper could be more clear. Reproducibility aspect of the paper looks weak.

---

> ### Author Response · Authors · 2022-11-16
> **Reply to Reviewer oFGQ**
>
> We thank the reviewer oFGQ's careful reading of the manuscript and constructive suggestions.
>
> **Q1: Readability.**
>
> We thank the reviewer’s careful reading of the manuscript. We tried to revise some expressions of our algorithm in the new manuscript. For example, we updated Algorithm 1 to improve readability.
>
> **Q2: Insufficient details or code available for reproducibility.**
>
> We have released the code in the supplementary material, and we believed it will help reproduce our results. We will add more details in the ReadMe file. We also added more descriptions in the experimental setting.

---

> ### Author Response · Authors · 2022-11-18
> **Available to discussion**
>
> Dear reviewer, do our responses address the concerns? We actually released our codes before the rebuttal. After the rebuttal, we tried to update the paper for improving its readability. Considering today, i.e., Nov. 18 is the deadline for the author's response, we hope our responses have addressed the reviewer's concerns, but if not we are available/open to address any outstanding issues. Please feel free to share your thoughts with us. We greatly appreciate your feedback or consideration for improving the score.

---

> > ### Comment · Reviewer_oFGQ · 2022-12-09
> > **Thanks but keeping the score unchanged**
> >
> > Thanks for the author response but after reading other reviews, I would keep my score unchanged

---

> > > ### Author Response · Authors · 2022-12-09
> > > **Thanks for your positive feedback and review**
> > >
> > > Dear Reviewer oFGQ,
> > >
> > > We sincerely appreciate your positive review and feedback with a score above acceptance.
> > >
> > > Bests,
> > >
> > > Paper6102 Authors.

---

### Official Review · Reviewer_LGnH · 2022-10-28

**Confidence:** 3
**Correctness:** 3
**Technical Novelty And Significance:** 2
**Empirical Novelty And Significance:** 2
**Recommendation:** 6

**Clarity, Quality, Novelty And Reproducibility:**

Section 3.2 could be made more clear. In particular, it's not very clear what \hat{x} is based on the text alone. Is it the example in the training set with the max confidence prediction for y*? Are these examples precomputed for a fixed clean model for each y*?

The accumulated gradient ranking and trojan weight pruning methods are technically simple. How are they different from existing bit flip methods? Is the main difference that existing methods only act on the last layer whereas these methods act on all parameters? The technical contribution compared to prior work should be made more clear.

There is sufficient detail for reproducibility.

**Strength And Weaknesses:**

Strengths:
- Memory editing attacks are an important threat model
- The proposed methods enable reaching higher ASR with a smaller decrease to clean accuracy
- A useful defense is proposed in Section 5.3 that seems to work well against the proposed bit-flip attacks

Weaknesses:
- I don't buy the distinction between training-time and test-time attacks. Surely if one can modify the model weights at test time via rowhammer to accomplish some goal, then this would be a form of training/tuning the network. It seems like the actual difference is that "test-time" or bit-flip attacks try to minimize the number of parameters that are changed, which makes sense for a rowhammer attack vector. This matters, because in the related work the authors mention that there are many new trojan detection techniques for text models (e.g., PICCOLO), and they claim that these detectors work much better on "training-time" attacks. However, I don't see any reason to believe this, since the authors do not include experiments with these detectors, and it's unclear that modifying fewer parameters results in a less detectable attack.
- Another issue with the test-time attack setting (or at least the version of it explored in this work) is that the trojan is inserted using data from the test set!!! This is akin to training on the test set, as it would artificially inflate ASR on the test set examples that were used for training the trojan. In reality, the victim model would be used on new samples from the data distribution that were not used for inserting the trojan, which is precisely why test sets are held out. (EDIT: One of the other reviewers pointed out that, based on the submitted code, it is likely that the authors used examples from the validation set. Could the authors confirm this?)
- In Algorithm 1, there are multiple epochs, and each epoch consists of editing all parameters with a distance greater than e to the corresponding benign model parameters. This doesn't place a constraint on the number of parameters edited, which seems more relevant for the memory editing attack vector. Also, the parameters that are edited could change in each epoch, which seems unrealistic. Wouldn't this allow you to arbitrarily change the entire model? It would be more realistic to fix the parameters that can be changed at the start of training and to not change them across epochs.


Other points:
- "trigger synthesizing is not applicable to textual model due to non-differentiable text tokens" There has been some work on trigger synthesis for textual trojans, e.g., the Neural Cleanse experiments in "Trojaning Language Models for Fun and Profit".
- RLI is called a contrastive loss, but it would be more accurate to call it an MSE loss, since it is just the MSE between two representations.
- The proposed defense is quite interesting. If this is novel, it should be expanded much more.

**Summary Of The Paper:**

This paper investigates bit-flip trojan attacks on text models with syntactic triggers. Several ideas are proposed for improving the effectiveness of these trojans. In experiments, this increases attack success rate (ASR) while having less of an effect on clean accuracy compared to baselines. Ablations confirm that modifying more bits leads to higher attack success rate. A defense is also proposed that can make models less susceptible to bit-flip attacks.

**Summary Of The Review:**

Given the weaknesses and various minor issues, I recommend to reject at this time.

----------------
Update:

The authors have addressed all of my concerns. I now think this is a good contribution to the line of work on bit flip attacks (especially because it focuses on the text domain and Transformers). I would give the paper a 7 if that was an option and will let the AC know.

---

> ### Author Response · Authors · 2022-11-16
> **Reply to Reviewer LGnH (1)**
>
> We thank reviewer LGnH for his/her careful reading of the manuscript and constructive comments.
>
> **Q1: [Important] The test-time attack setting. Make sure the training and testing datasets are not the same.**
>
> The reviewer YubY also asked this question and we copy the answer here for convenience.
>
> We insert Trojans into victim models on the validation dataset. And we test the model, e.g., ASR, on the test dataset, thus we DO NOT train and test on the same split dataset, e.g., the same test dataset. There is no overlap between validation and test datasets. In our setting, we assume that the training dataset and testing dataset are unavailable for attackers, but the attackers can obtain shadow test datasets (NOT test datasets), i.e., validation datasets for training. In this way, attackers can use the validation dataset to generate the poisoned dataset with the syntactic trigger. Then, attackers combine the clean and poisoned test datasets together and feed them to the target model to poison the target model. After training, victim users will use a real test dataset to test the performance of our attack. A clarification about the testing attacks is updated in Section 4.1.
>
> **Q2:  Difference between training-time and test-time attacks? It seems like the actual difference is that "test-time" attacks try to minimize the number of parameters that are changed.**
>
> The difference between “training-time” and “testing-time” attacks is NOT on the number of parameters to be changed. Their difference is mainly in the time to insert the Trojan attacks. In particular, the pipeline of deep learning can be basically divided into two phases: 1. training and 2. testing. During the phase of training, the model is trained on the training dataset. After the training, the model is distributed even to other parties and devices for deployment. During the testing phase, a machine runs the trained model for testing. Training-time attacks mean that Trojans are inserted during the training phase and dependent on training data. In contrast, testing-time attacks mean that Trojans are inserted during the testing phase without reliance on training data.
>
> **Q3: Why do detectors like PICCOLO work much better on "training-time" attacks than test-time attacks? Is it because testing-time attacks modified fewer parameters?**
>
> Detectors like PICCOLO work much better on "training-time" attacks because these detectors try to detect if a given trained model is Trojaned or not. However, a testing-time attack inserts the Trojan after the detection and deployment, thus might bypassing the pre-deployment detection like PICCOLO. We argue that this potential advantage of testing-time attacks is not first claimed by our work. Previous test-time attacks like TBT [1] and Proflip[2] also claimed this. Our contribution is NOT to present a test-time attack. Instead, we follow the previous test-time attacks to design a more feasible and practical invisible Trojan attack without a training dataset.
>
> **Q4: Unclear algorithm 1.**
>
> We only set one-layer weights of the target model as trainable, and do not update all the parameters. Also, for every epoch, only important parameters (top k important parameters selected by Accumulated Gradient Ranking) in the chosen layer are updated and the other parameters will be restored back to their original values.
>
> We update our algorithm 1 to add more implementation details in the manuscript. In particular, we define an initialized variable, denoted by index, outside of for loops to represent all parameters’ indices that we could change.  For each epoch, we prune the index_p whose parameters have a distance less than e over the original parameters of clean models. Then we iteratively remove the index_p from the index to reduce the parameter number to be changed. Our released codes are also helpful to reproduce our algorithm 1.
>
>
> **Q5: Unclear descriptions. The claim about “Such Trigger synthesizing is not applicable to textual model due to non-differentiable text tokens" is not proper since there has been some work on trigger synthesis for textual trojans.**
>
> We did NOT claim that trigger synthesis for textual trojans is not applicable. We claimed that directly transferring the gradient-based trigger synthesizing in computer vision to the text domain is not applicable due to non-differentiable text tokens. We elaborated on this point and updated our paper to avoid confusion.
>
> **Q6. RLI is called a contrastive loss, but it would be more accurate to call it an MSE loss.**
>
> As Section 3.2 shows, RLI is NOT an MSE loss between two representations, instead RLI loss is a loss function that combines the loss of classification logits ($L_L$) and the loss of encoding representation ($L_R$). $L_R$ is actually a contrastive representation loss implemented by MSE between two representations.

---

> > ### Author Response · Authors · 2022-11-16
> > **Reply to Reviewer LGnH (2)**
> >
> > **Q7. The proposed defense discussion**
> >
> > We used the existing matrix decomposition method as one potential defense method against our attacks. The idea to use this matrix decomposition is interesting and we provided results and implementation for reproducibility and hope to inspire more following defense methods.
> >
> > **Q8: Section 3.2 could be made more clear. In particular, it's not very clear what $\hat{x}$ is based on the text alone. Is it the example in the training set with the max confidence prediction for $y^\*$? Are these examples precomputed for a fixed clean model for each $y^\*$?**
> >
> > $\hat{x}$ is selected from clean sentences. For target label sentences, we input them into a fixed clean target model to precompute the confidence score for $y^*$ and select the sentence with the max confidence score as the representative target sentence $\hat{x}$.
> >
> > **Q9: Contribution and novelty clarification.**
> >
> > Existing bit-flip attacks all focus on convolutional neural networks in the domain of computer vision. Our work is the first bit-flip attack focusing on an attention-based Transformer in the domain of natural language processing. It is not trivial or straightforward to achieve an accurate and feasible invisible bit-flip attack for a text Transformer.
> >
> > Since no existing bit-flip attacks on text Transformer, we first construct our baseline that performs syntax trigger attack on Transformer, e.g., Hidden Killer [3] with an existing bit-flip method, e.g., TBT [1]. We then compare our methods with our baseline from Table 2 to Table 5.  Compared to the baseline model, our method improves the CACC and ASR by 2.53% and 3.66% on average. Importantly, our method decreases the bit-flip numbers by 51.59% on average.
> >
> > Our baseline only uses the logits loss, however, our method proposes representation and logits loss, denoted representation-logit Trojan Insertion (RLI).  We identified that when the training dataset is not available, only using the logits loss to train the model on a small dataset, e.g., a validation dataset, is difficult to get convergent and high ASR and CACC. Compared to previous logits-based loss in our baseline, our RLI has additional optimization on the encoding representations, thus providing one more constraint on the weights optimization space, which is very helpful when training data is not substantial. For example, Table 2 in the new manuscript shows that our TrojText-R achieves a higher 6.29% ASR and 1.02% CACC over our baseline.
> >
> > Our baseline uses Neural Gradient Ranking (NGR) on the last layer to rank the top k important parameters based on one random batch of data. We identify one key problem of NGR: the top-k important parameters on one random batch of data are not always important for different batches of data. The importance of parameters varies significantly over different batches of data. For this reason, NGR based on one random batch of data is not stable or accurate across different batches of datasets, which will decrease the stability and accuracy of the attack significantly. We propose Accumulated Gradient Ranking (AGR) to tackle the instability and inaccurate issue. AGR selects top-k important parameters on all batches of the dataset by computing the accumulated gradients across all batches. Therefore, AGR stably and correctly identifies important parameters for attacks. For example, Table 2 in the new manuscript shows that our TrojText-RA achieves 6.29% higher ASR and 1.02% higher CACC over our TrojText-R. Note TrojText-RA is constructed by adding AGR on TrojText-R.
> > Since the bit-flip number is very important to the attack cost. We further propose Trojan Weights Pruning to reduce the bit-flip number. The details are shown in Algorithm 1. Compared to TrojText-RA, the TrojText-RAT reduces almost 50\% bit-flip numbers shown in Table 2.
> >
> > [1]Rakin, A. S., He, Z., & Fan, D. (2020). TBT: Targeted Neural Network Attack With Bit Trojan. Proceedings of the IEEE/CVF Conference on Computer Vision and Pattern Recognition (CVPR).
> >
> > [2] Chen, H., Fu, C., Zhao, J., & Koushanfar, F. (2021). ProFlip: Targeted Trojan Attack With Progressive Bit Flips. Proceedings of the IEEE/CVF International Conference on Computer Vision (ICCV), 7718–7727.
> >
> > [3] Qi, F., Li, M., Chen, Y., Zhang, Z., Liu, Z., Wang, Y., & Sun, M. (2021). Hidden killer: Invisible textual backdoor attacks with syntactic trigger. arXiv preprint arXiv:2105.12400.

---

> > ### Comment · Reviewer_LGnH · 2022-12-06
> > **Thanks for the clarifications!**
> >
> > > for every epoch, only important parameters (top k important parameters selected by Accumulated Gradient Ranking) in the chosen layer are updated and the other parameters will be restored back to their original values.
> >
> > Thanks for the clarification. Just to help me understand, does this mean that after training only a small set of parameters are changed, which you can then target with the actual rowhammer attack?
> >
> >
> > > However, a testing-time attack inserts the Trojan after the detection and deployment, thus might bypassing the pre-deployment detection like PICCOLO.
> >
> > Calling PICCOLO a pre-deployment detector seems like an artificial categorization. Surely if a defender is worried about rowhammer attacks modifying the parameters of their model, they could run these detectors on a regular basis. I think there are very good reasons to evaluate detectors like PICCOLO. Just because TBT and Proflip didn't run these kinds of detectors, that doesn't mean you shouldn't. If the results of the experiments are that bit-flip attacks are easy to detect, then that would be valuable for the community to know (imo it would strengthen the paper relative to prior work like TBT and Proflip if they ignored this fact).
> >
> > > L_R is actually a contrastive representation loss implemented by MSE between two representations.
> >
> > This is a minor point, but people have been using the term "contrastive loss" in an imprecise manner. I'm pretty sure it used to refer specifically to triplet losses, but it has since taken on a broader meaning. I was just pointing out that the authors aren't using the term in a precise sense, and it would improve clarity to call L_R an MSE loss.

---

> > > ### Author Response · Authors · 2022-12-06
> > > **Thanks and rebuttal to Reviewer LGnH's follow-up questions**
> > >
> > > We remain indebted to your valuable comments that have helped us clearly demonstrate the true value. We remain committed to clarifying your follow-up questions associated with the initial rejection and hope to receive your support in this scientific endeavor.
> > >
> > > **New Q1: Does this mean that after training only a small set of parameters are changed, which you can then target with the actual rowhammer attack?**
> > >
> > > We thank reviewer LGnH for his/her efforts in identifying valuable points of this paper. Yes, reducing the weight parameter modification number is crucial for actual and real-world rowhammer attacks, which is why we managed to make sure a small set of parameters are changed after training.
> > >
> > > **New Q2: Calling PICCOLO a pre-deployment detector seems like an artificial categorization. Surely if a defender is worried about rowhammer attacks modifying the parameters of their model, they could run these detectors on a regular basis. I think there are very good reasons to evaluate detectors like PICCOLO. Just because TBT and Proflip didn't run these kinds of detectors, that doesn't mean you shouldn't. If the results of the experiments are that bit-flip attacks are easy to detect, then that would be valuable for the community to know (imo it would strengthen the paper relative to prior work like TBT and Proflip if they ignored this fact).**
> > >
> > > We understand the reviewer's doubt and apologize for this confusion. Figure 1 in the PICCOLO paper shows that PICCOLO is originally dedicated to the pre-deployment backdoor scanning method since PICCOLO is performed to scan a pre-trained model that is downloaded from the cloud.  If a defender is worried about rowhammer attacks, a backdoor scanning method, e.g., trigger-inversion-based PICCOLO is NOT an ideal solution due to a large scanning overhead although they can accurately detect the backdoor attacks. We summarize the inefficiency reasons as follows:
> > >
> > > Firstly, compared to model inference time, PICCOLO scanning time is huge, i.e., it is similar to a small finetuning to inversely generate triggers.  As PICCOLO shows, each BERT inference takes less than 0.1 seconds but one scanning takes almost 350 seconds, thus the scanning will reduce the system efficiency, especially for models deployed on personal and mobile devices.
> > >
> > > Secondly, multiple and periodical scanning is required for rowhammer attack detection.  This is because those rowhammer attacks are dynamic by modifying the model weights in volatile DRAM, e.g., the rowhammer attacks may be inserted after a long time running the program. Therefore, it is difficult for users or defenders to know the time the model is backdoored and the time to insert the scanning. If only one scanning is performed, the detection may ignore the backdoor that is injected after the scanning. One solution is to periodically scan the model, which is, however, time-consuming since multiple scanning is required.
> > >
> > > However, it is more efficient and feasible to adopt our proposed defense method shown in our manuscript. Simply decomposing the target model’s weight will bring challenges for attackers to find corresponding important parameters, thus decreasing the attack success rate.  The potential defense method in our manuscript is a low-latency and energy-efficient method because the users/defenders only need to decompose the model once before the deployment.
> > >
> > > **New Q3: This is a minor point, but people have been using the term "contrastive loss" in an imprecise manner. I'm pretty sure it used to refer specifically to triplet losses, but it has since taken on a broader meaning. I was just pointing out that the authors aren't using the term in a precise sense, and it would improve clarity to call L_R an MSE loss.**
> > >
> > > We would like to express our sincere thanks for this suggestion. Our L_R is originally designed to contrast two representations and can be implemented by an MSE loss. We will follow the reviewer's suggestion and appreciate the support.

---

> > > > ### Comment · Reviewer_LGnH · 2022-12-11
> > > > **Questions about detection**
> > > >
> > > > > As PICCOLO shows, each BERT inference takes less than 0.1 seconds but one scanning takes almost 350 seconds, thus the scanning will reduce the system efficiency, especially for models deployed on personal and mobile devices.
> > > >
> > > > The cost would be low if one is amortizing, such that detection is only run every now and then, or (in the case of mobile) when the device is idling so that the user wouldn't notice. Unless one is a high-profile target, it seems reasonable that scanning for rowhammer attacks on mobile devices would only need to be done, e.g., once per day.
> > > >
> > > > More importantly, running model-level detectors on these attacks seems scientifically valuable. I think many readers would be interested in how existing detection methods perform against bit flip attacks. PICCOLO's code is readily available on GitHub and can be run out-of-the-box with minimal effort, so I think it makes sense to run it on the proposed attack and report numbers. I understand that people haven't done this for bit flip attacks before. This is exactly what makes it so valuable for the community.
> > > >
> > > > > it is more efficient and feasible to adopt our proposed defense method shown in our manuscript
> > > >
> > > > As I mentioned in my original review, I think the proposed defense is interesting, but the authors should include more details. Which decomposition is used, and how is it applied? If the decomposition is the SVD, and the original weight matrices are just swapped out with their corresponding U, S, and V components, then wouldn't an attacker be able to make a huge change by modifying the top-left element in S?
> > > >
> > > > Also, if the defense involves decomposing each W into a USV multiplication, wouldn't that approximately double the number of sequential operations in each forward pass?

---

> > > > > ### Author Response · Authors · 2022-12-11
> > > > > **Responses to the follow-up detection questions**
> > > > >
> > > > > **Q1: The cost would be low if one is amortizing, e.g., such that detection is only run when the device is idling so that the user wouldn't notice.**
> > > > >
> > > > > Thanks for your question. Amortizing detection time to hide overhead is a potential improvement. However, it seems that energy efficiency is not solved by this amortizing method and an efficient system in the real world requires both latency efficiency and **energy efficiency**.  A detection that takes 350 seconds on a \~750 Watts GPU consumes \~4.4 KWh of energy. Mobile phones, e.g., the iPhone that only has at most 5.5 KWh \(Apple reports that the iPhone 13 with running power of 0.1~0.5 watts can hold for ~10 hours\), will consume 4.4/5.5=80\% battery energy to run one detection after one-time full charging. Performing detection during charging may be one solution, however, it has a dark-side effect on reducing the battery life due to the limited charging number. Thus, it is open but still valuable and non-trivial to think about how to modify the existing detection methods like POCOLLO to develop a more efficient \(latency, energy, carbon footprint\) on-device detection method for rowhammer attacks.
> > > > >
> > > > > **Q2: Running model-level detectors on these attacks seems scientifically valuable.**
> > > > >
> > > > > We totally agree that it is valuable to run model-level detectors on test-time row hammer attacks.  We also had the same questions with the reviewer about why previous row hammer attacks including TBT and ProFlip did not run these detections. We found that there are two reasons, the first is that the model-based model-level detectors may be bypassed if defenders use it once before deployment, which is supported by previous rowhammer attacks like TBT and Proflip; the second is an efficiency issue although it is effective to detect the rowhammer attacks. We show the detection effectiveness in the following table.  Here we run POCOLLO using BERT on SSL-2 and AG's-News datasets. Columns TP, FP, FN, and TN denote the number of true positives, false positives, false negatives, and true negatives, respectively. Column Accuracy presents the overall detection accuracy.
> > > > >
> > > > > | Dataset             | TP        |FP           | FN               | TN                | Accuracy |
> > > > > |---------------------|-------------|-------------|------------------|:----------------:|:----------------:|
> > > > > | SSL-2             | 18     | 1       | 2          |19         |0.925|
> > > > > | AG's-News    | 20       | 0       | 0            | 20            | 1.0             |
> > > > >
> > > > > Although current model-scanning methods are effective to detect, we think it is valuable, non-trivial, and open to considering how to modify the existing detection methods like POCOLLO to enable a more efficient \(latency, energy, carbon footprint\) on-device detection method. We argue that our works mainly focus on improving the efficiency and feasibility of row hammer attacks for invisible triggers and detection improvement. We would appreciate it very much if the reviewer can give us support on the proposed attacking methods, especially since we carefully conduct experiments and make ablation studies on each proposed method on multiple datasets and various networks.
> > > > >
> > > > > **Q3: Clarify more potential defenses, e.g., process and cost of forward inference**
> > > > >
> > > > > Thanks for the questions. We used a model with weight decomposition instead of the original model as a potential defense method. The decomposition method based on SVD factorization with half rank we cited in the paper is widely used in model compression and inference speedup without a large model performance decrease. Thus the model forward cost is not increased.
> > > > >
> > > > > Why does decomposition work? If we decompose θ  in Equation 5 to three components, the proposed accumulated gradient ranking method used by attackers cannot work correctly to identify the critical weights as before. This is one reason why our work has defense effects. We argued that this defense method is only for our specific method using Equation 5. If attackers do not use Equation 5 or use other attack methods, this decomposition method may not work and may even be much easier for attacking. We argue that our works mainly focus on improving the efficiency and feasibility of row hammer attacks for invisible triggers. How to efficiently detect and mitigate the attacks is still one open question. We would appreciate it very much if the reviewer can give us support on the proposed attacking methods, especially since we carefully conduct experiments and make ablation studies on each proposed method on multiple datasets and various networks.
> > > > >
> > > > >
> > > > > ​

---

> > > > > > ### Comment · Reviewer_LGnH · 2022-12-11
> > > > > > **Response**
> > > > > >
> > > > > > > We totally agree that it is valuable to run model-level detectors on test-time row hammer attacks. We also had the same questions with the reviewer about why previous row hammer attacks including TBT and ProFlip did not run these detections.
> > > > > >
> > > > > > Thank you for running these experiments. The results are surprising, because one wouldn't expect a small number of bit flips to be detectable. I don't think this is an issue for the proposed attack, because defenders might not think to run a backdoor detector on their networks after the fact, as the authors mention. The authors could also probably change the trigger to something different and possibly disrupt PICCOLO. The ability to insert trojans in this setting is the important thing, and this is an additional experiment that strengthens the paper by clarifying the threat landscape.
> > > > > >
> > > > > >
> > > > > > > Why does decomposition work?...
> > > > > >
> > > > > > I was confused earlier; modifying the largest singular value would just mess up the network; it wouldn't be a good way to insert a trojan. Thanks for clarifying that the inference time doesn't increase.

---

> > > > > > > ### Author Response · Authors · 2022-12-11
> > > > > > > **Thanks for support and feedback**
> > > > > > >
> > > > > > > We appreciate it very much for your constructive feedback and review. And it is good to know that our rebuttals help resolve the concerns.  We will explore the possibilities to disrupt PICCOLO detection with different invisible triggers for backdoor attacks based on row hammers.
> > > > > > >
> > > > > > > Also, we would like to express our sincere thanks for the increased score and support.

---

> ### Author Response · Authors · 2022-11-18
> **Follow-up discussion**
>
> Dear reviewer, do our responses address the concerns? Please feel free to share your thoughts with us. We greatly appreciate your feedback. Considering today, i.e., Nov. 18 is the deadline for the author's response, we hope our responses have addressed the reviewer's concerns, but if not we are available/open to address any outstanding issues. Thank you very much.

---

> ### Author Response · Authors · 2022-11-24
> **Available for Discussion**
>
> We hope our responses have addressed the reviewer's concerns, but if not we are available/open to address any outstanding issues.

---

### Decision · Program_Chairs · 2023-01-20

**Decision:**

Accept: poster

**Justification For Why Not Higher Score:**

Although the paper meets the bar, reviewers are not excited about the paper. The proposed approach is relatively incremental and contribution is relatively narrow.

**Justification For Why Not Lower Score:**

There is no major concern about the paper.

**Metareview: Summary, Strengths And Weaknesses:**

The paper studies Text Trojan attacks and investigates whether the invisible Trojan attacks can be done without knowing training data. The paper further proposed a representation-logic Trojan insertion approach and methods to reduce parameter tuning. Overall, there is no major concern about the paper; however, the paper remains on the borderline as the novelty and significance are moderate and the writing can be further improved.

Strengths:
-	The paper studies an interesting problem of backdoor insertion attacks.
-	The experiments sufficiently support the paper.
-	There is no major concern after the rebuttal discussion

Weaknesses:
-Although the authors resolve some writing issues and clarify the confusion in the rebuttal, the overall writing of the paper can be improved. There are still several places remaining unclear and need careful revision.
- While the paper argues that their setting is realistic, it should admit the limitations of testing the approaches on AG's News, SST-2, and OLID. Although I understand that these datasets have been used as benchmarks for adversarial text attack over years, the motivation of attacking these applications is weak and there is no real-world applications/scenarios associated with attacking sentiment analysis and news category classifier. I would recommend that the papers consider applications such as toxicity classifiers or spam detection models.


**Note From Pc:**

if the above contains the word "oral" or "spotlight" please see: "oral" presentation means -> notable-top-5% and "spotlight" means -> notable-top-25%. As stated in our emails, we are disassociating presentation type from AC recommendations

**Summary Of Ac-Reviewer Meeting:**

The AC has attempted to schedule the meetings through several channels, but only reviewers YubY and LGnH respond to the meeting requests. Unfortuantely, due to the time difference and extensive conference travel schedule of AC and reviewers, the AC only managed to meet reviewer YubY in person at EMNLP. However, the AC went over the reviews with YubY and concluded that the reviews were consistent in ratings and comments. Some other reviewers also provide their comments offline.

Overall, there is no harm to accept this paper, although reviewers are not excited about this paper as well.